# Precise Targeting of Autoantigen-Specific B Cells in Lupus Nephritis with Chimeric Autoantibody Receptor T Cells

**DOI:** 10.3390/ijms25084226

**Published:** 2024-04-11

**Authors:** Cristina Solé, Maria Royo, Sebastian Sandoval, Teresa Moliné, Alejandra Gabaldón, Josefina Cortés-Hernández

**Affiliations:** 1Rheumatology Research Group, Lupus Unit, Hospital Universitari Vall d’Hebron, Institut de Recerca (VHIR), Universitat Autònoma de Barcelona, 08035 Barcelona, Spain; maria.royo@vhir.org (M.R.); sebastian.sandoval@vhir.org (S.S.); fina.cortes@vhir.org (J.C.-H.); 2Department of Pathology, Hospital Universitari Vall d’Hebron, Institut de Recerca (VHIR), Universitat Autònoma de Barcelona, 08035 Barcelona, Spain; teresa.moline@vhir.org (T.M.); magabaldon@vhebron.net (A.G.)

**Keywords:** lupus nephritis, CAART, anti-dsDNA, B cell

## Abstract

Despite conventional therapy, lupus nephritis (LN) remains a significant contributor to short- and long-term morbidity and mortality. B cell abnormalities and the production of autoantibodies against nuclear complexes like anti-dsDNA are recognised as key players in the pathogenesis of LN. To address the challenges of chronic immunosuppression associated with current therapies, we have engineered T cells to express chimeric autoantibody receptors (DNA-CAART) for the precise targeting of B cells expressing anti-dsDNA autoantibodies. T cells from LN patients were transduced using six different CAAR vectors based on their antigen specificity, including alpha-actinin, histone-1, heparan sulphate, or C1q. The cytotoxicity, cytokine production, and cell–cell contact of DNA-CAART were thoroughly investigated in co-culture experiments with B cells isolated from patients, both with and without anti-dsDNA positivity. The therapeutic effects were further evaluated using an in vitro immune kidney LN organoid. Among the six proposed DNA-CAART, DNA4 and DNA6 demonstrated superior selectively cytotoxic activity against anti-dsDNA^+^ B cells. Notably, DNA4-CAART exhibited improvements in organoid morphology, apoptosis, and the inflammatory process in the presence of IFNα-stimulated anti-dsDNA^+^ B cells. Based on these findings, DNA4-CAART emerge as promising candidates for modulating autoimmunity and represent a novel approach for the treatment of LN.

## 1. Introduction

Systemic lupus erythematous (SLE) is a chronic inflammatory autoimmune disease with a broad spectrum of manifestations and organ involvement [1,2]. Lupus nephritis (LN) occurs in 40–50% of SLE patients and is associated with an unpredictable course. Despite conventional therapy, LN is still a major cause of short- and long-term morbidity and mortality [3,4], with up to 20% of patients progressing to an end-stage renal disease (ESRD) [4].

LN etiopathogenesis is complex and multifactorial. It is characterised by a loss of tolerance to endogenous nuclear antigens, polyclonal B-cell hyper-reactivity, the production of multiple antibodies, and immune-mediated injury to the kidney, resulting in tissue inflammation/damage if left untreated [5,6]. B cell abnormalities play a pivotal role in the pathogenesis of SLE and LN [7,8,9]. A breach to central and peripheral tolerance mechanisms generates autoreactive B cells, which contribute to pathogenesis [10]. The dysregulation of B cell transcription factors, cytokines, and B cell–T cell interaction can result in aberrant B cell maturation, autoantibody production, and inflammation [9,10]. Corticosteroids, antimalarials, and immunosuppressive agents remain as first-line therapies and confer differential effects to B cells [2,11]. Novel therapeutic approaches targeting the B cell repertoire include B cell depletion with monoclonal antibodies against cell surface antigens (anti-CD20), the inhibition of B cell cytokines or survival factors (anti-BLyS or belimumab), and the modulation of costimulatory signals in B cell–T cell interaction [12]. Even though results from randomised controlled trials (RCTs) (BLISS-52 and BLISS-76, BLISS-LN) showed that belimumab treatment had moderate efficacy in active SLE and LN patients [13,14,15], the efficacy of anti-CD20 antibody therapy is controversial. Rituximab has demonstrated its efficacy as an add-on therapy to standard treatments in refractory LN and severe non-renal SLE manifestations [16,17], but RCTs have failed to demonstrate its efficacy, despite improvements in serological parameters. Failure may be attributed to the transient and incomplete B cell depletion both in peripheral blood and at a renal tissue level.

Chimeric antigen receptor (CAR) technology has revolutionised cancer therapy, causing specific and permanent B cell depletion leading to durable disease remission [18,19]. CAR-T cells have been tested in a few autoimmune diseases [20]. In SLE, animal studies provide support for the potential efficacy of CD19 CAR-T cell therapy [21,22]. Furthermore, initial human studies reinforce this notion by inducing persistent CD19 B cell depletion, a reduction in autoantibody production, as well as clinical remission [23,24]. The main advantages over cell-based therapies are single administration, the ability to migrate to multiple lymphoid tissues and organs, and transformation into both effector and memory cell populations. However, therapy is not selective and can have life-threatening side effects. In autoimmune diseases, chimeric autoantibody receptor T (CAART) therapy, selectively targeting antibody-producing autoreactive B cells, could be an alternative approach [25]. Preclinical and clinical studies have shown the efficacy of this approach in pemphigus vulgaris [26], MuSK autoantibody-positive myasthenia gravis [27], and NMDA receptor antibody encephalitis [28]. Recently, an in vitro study showed that myelin basic protein (MBP)-CAART has higher cytotoxic activity against autoreactive B cells in experimental autoimmune encephalomyelitis (EAE) [29].

Several autoantibodies, especially those against double-stranded DNA (anti-dsDNA), are believed to play a major role in the induction of glomerular inflammation [30,31,32,33]. Antibodies to dsDNA are specific to SLE and are detected in 70–80% of LN patients [34,35], while in the remaining LN cases, levels are undetectable (5–30%) [36]. Serum anti-dsDNA titres correlate with disease activity, can precede renal manifestations, and their immune complexes have been detected in the glomerulus [37]. In addition to forming immune complexes and triggering complement activation, anti-dsDNA antibodies contribute to LN through binding directly or indirectly to cross-reactive antigens on the surface of resident renal cells, components of the extracellular matrix, or chromatin materials, thereby triggering the downstream cellular activation of signalling pathways and the release of mediators of inflammation and fibrosis [38,39,40,41].

We reasoned that by expressing specific antigens as the extracellular domain of a chimeric immunoreceptor, cytotoxicity would become specific for only those B cells bearing anti-specific antigens, providing targeted therapy for LN without general immunosuppression. As an alternative to using DNA as an antigen, we designed a panel of specific LN antigens known to cross-react with anti-DNA antibodies. We used heparan sulphate, alpha-actinin, histone 1, and C1q as the CAAR extracellular domain to engineer T cells and select, in vitro, the best CAAR to kill specifically autoimmune anti-dsDNA producer B cells in LN.

## 2. Results

### 2.1. Patient Demographic Characteristics

Clinical characteristics based on patients’ anti-dsDNA positivity are shown in Table 1. There were no significant differences between the two groups with respect to demographic and disease characteristics. All patients had a previous history of type III/IV proliferative GMN. Only two had concomitant type V GMN. Both groups had persistent anti-DNA positivity or negativity for at least 3.7 years since renal flare (range 2.5–4.9 years). Patients with persistent anti-dsDNA positivity had more anti-C1q (60%, *p* = 0.011), anti-histone1 (50%, *p* = 0.05), anti-α-actinin (60%, *p* = 0.05), and anti-heparan sulphate antibodies (90%, *p* = 0.001) at inclusion compared with the anti-dsDNA^−^ patients. At the time of renal biopsy, the antibody profile was similar between both groups except for the titre of anti-dsDNA antibodies, which was higher in the group with persistent anti-dsDNA positivity (*p* = 0.073). No differences were observed regarding the presence of anti-SSA/Ro, anti-SSB/La, anti-RNP, or anti-Sm.

### 2.2. Design and Construction of Six DNA-CAART

We aimed to engineer DNA chimeric autoantibody receptor DNA-CAART cells expressing an antigen on their surface able to eliminate anti-dsDNA-producing B cells. Several antigens, frequently those expressed in renal tissue, exhibit cross-reactivity with circulating anti-dsDNA [41]. Of those, the most cross-reactive antigens include α–actinin, heparan sulphate, histone-1, and C1q. Considering this, we proposed and designed six chimeric autoantibody receptors (DNA-CAAR) with autoantigen protein sequences of α-actinin (n = 2, DNA1 or DNA2-CAAR), heparan sulphate (n = 2, DNA3 or DNA4-CAAR), histone-1 (n = 1, DNA5-CAAR), and C1q protein (n = 1, DNA6-CAAR) (Figure 1A).

DNA1-, DNA2-, DNA3-, DNA4-, and DNA6-CAART showed comparable transfection efficacy in T cells from anti-dsDNA^+^ LN patients (66.0% ± 5.4, 66.0% ± 4.4, 65.1% ± 4.7, 61.6% ± 4.2, 64.6% ± 5.5, respectively, Figure 1B), whereas DNA5-CAART expression was significantly lower (13.5% ± 5.7%, Figure 1B). In addition, DNA1-4 and DNA6-CAART exhibited significantly higher expression of their corresponding antigen genes compared to DNA5-CAART (fold change over non-transduced T cells; fold of 20.0 and 23.1 for α-actinin; fold of 26.1 and 25.05 for heparan sulphate; fold of 25.6 for C1q vs. fold of 1.5 for histone-1, Figure 1C). Similar results were observed using T cells from anti-dsDNA^−^ LN and healthy groups, indicating that transduction efficiency is not sample source-dependent (Appendix A).

### 2.3. DNA4 and DNA6-CAART Show Higher Selective Depletion of Anti-dsDNA^+^ B Cells

We evaluated the ability of DNA1-4 and DNA6-CAAR T cells to kill anti-dsDNA-producing B cells in vitro. DNA5-CAART was discarded from further experiments due to its low transfection efficacy in primary T cells. First, we evaluated the capacity of B cells isolated from anti-dsDNA^+^ LN patients to produce anti-dsDNA antibodies. Cells were treated with IFNα, IFNβ stimulation, or non-stimulation conditions for 48 h. IFNα and IFNβ stimulation produced increased titres of anti-dsDNA antibodies compared to non-stimulatory conditions after 24 h. This increment was higher and sustained over time (*p* < 0.001, Figure 2A) following IFNα stimulation. B cells from anti-dsDNA^−^ LN patients also produced anti-dsDNA antibodies following IFNα stimulation, but at a lower level, when compared with anti-dsDNA^+^ LN patients (187 ± 134 vs. 4909 ± 1579 UI/mL, Figure 2B). On the contrary, there was no antibody production in non-stimulated or IFNβ-stimulated B cells isolated from anti-DNA^−^ LN patients or healthy donors (Appendix A).

Next, to evaluate cytotoxicity towards primary human anti-DNA-producing B cells, we co-incubated non-transduced (NTD, control) and DNA1-4 or DNA6-CAART with IFNα-stimulated primary B cells from anti-dsDNA^+/−^ patients or healthy donors for 48 h using several effector–target cell ratios (E:T, Figure 2C). At 24 h, DNA4 and DNA6-CAARTs with an E:T ratio of 10:1 showed the highest specific lysis efficacy for anti-dsDNA^+^ B cells (cytotoxicity of 82.5% and 78.9%, respectively) and no cytotoxicity for anti-dsDNA^−^ B cells (Figure 2C). DNA1-CAART showed lower cytotoxicity for anti-dsDNA^+^ B cells (E:T of 10:1, cytotoxicity of 43.5%) and significantly similar cytotoxicity for anti-dsDNA^−^ B cells, showing non-specific lysis (cytotoxicity of 36.2%, *p* < 0.001, Figure 2C). DNA2 and DNA3-CAART showed less than 10% B cell cytotoxicity after 24 h of co-incubation (Figure 2C).

Likewise, RNA expression analysis showed a high expression of apoptotic genes (caspase 3, BIM, and p53) in anti-DNA^+^ B cells when co-incubated with DNA4-CAART (fold changes of 117.5, 20.1, and 12.1, respectively) and with DNA6-CAART (fold changes of 45.5, 21.7, and 13.4, respectively) compared to anti-dsDNA^−^ B cells, suggesting a higher cytotoxic effect (Figure 3A). Enzyme-linked immunosorbent assay (ELISA) also revealed a substantial IFNγ secretion in the culture medium by the DNA4 and DNA6-CAART after their co-culture with anti-dsDNA^+^ B cells, as well as a reduction in anti-dsDNA titres (Figure 3B,C). No significant production of IFNγ or antidsDNA reduction was observed following co-culture with anti-dsDNA^−^ B cells (Appendix A).

To determine soluble anti-dsDNA effects on DNA4 and DNA6-CAART activity, we performed cytotoxicity assays adding standard of an anti-dsDNA (0, 50, 100, 250, 500 UI/mL) into the co-culture with anti-dsDNA^+^ B cells, at concentrations within or exceeding the expected range. We observed that cytotoxicity remained consistent, with values ranging between 80 and 70%, in the presence of free soluble anti-dsDNA antibodies (0–500 UI/mL, Figure 3D). However, when the concentration was 1000 or 2000 UI/mL, above the expected range, we observed decreases in cytotoxicity to 44–35% and 25–15%, respectively (Appendix A).

### 2.4. Confocal Live-Cell Imaging Provides Visualization of DNA4 and DNA6-CAART Interaction and Expansion with Anti-dsDNA^+^ B Cells

We used immunofluorescence to further investigate the interaction between DNA-CAART and target B cells. Over 24 h, we co-cultured DNA4 or DNA6-CAART with PE-CD19 labelled B cells isolated from anti-dsDNA^+^ or anti-dsDNA^−^ LN patients in a 10:1 (E:T) ratio. There was a significant reduction in PE-CD19 B cells from anti-dsDNA^+^ LN patients that had been co-incubated with the DNA4-CAART compared with those from anti-dsDNA^−^ LN (76% vs. 6% reduction in PE-CD19 B cells, Figure 4A). DNA6-CAART also produced a significant reduction in PE-CD19 B cells from anti-dsDNA^+^ LN patients (67% vs. 15%, *p* < 0.001, Figure 4A).

Using confocal live-cell imaging, we visualised DNA4 and DNA6-CAART engagement with anti-dsDNA^+^ B cells (Figure 4B). Time-lapse microscopy showed a differential kinetic curve between DNA4-CAART and DNA6-CAART. Significant differences at 2 h and 6 h were found between DNA4 and DNA6-CAART (*p* = 0.05 and *p* = 0.008, Figure 4B). While DNA4-CAART reached the major cell–cell contact with anti-dsDNA^+^ B cells at 2 h, DNA6-CAART reached the peak of contact interaction at 6 h (Figure 4B), indicating a quicker interaction between effector and target cells in DNA4-CAART.

### 2.5. Therapeutic Evaluation of DNA4 and DNA6-CAART Cells in 3D Immuno-Kidney Model Assay

We designed an organoid-immune co-culture using an established human kidney organoid (provided by HubOrganoid Company) and isolated B cells from anti-dsDNA^+^ LN patients (Figure 4A). The kidney organoids were completely characterized by Schutgens et al. [43]. Kidney organoids were cultured in a 24-well plate for 5 days using Matrigel to achieve 70% confluence. Meanwhile, B cells from anti-dsDNA^+^ LN patients were stimulated with IFNα and injected directly into the Matrigel where the kidney organoid (1 × 10^3^ cells in each well, Figure 5A) was suspended.

After 2 days of co-culture, kidney organoids were significantly smaller and partially disintegrated compared to controls (circularity 0.057 vs. 0.981; roundness 0.743 vs. 0.899; aspect ratio 1.30 vs. 1.05; and solidarity 0.760 vs. 0.975, respectively, Figure 5B. As additional controls, we utilized IFNα-stimulated B cells from healthy donors and from anti-dsDNA^−^ LN patients. We observed slightly significant changes in the circularity but not in the other morphology parameters of the kidney organoids (Appendix A). We used confocal microscopy to further characterize the interaction between IFNα-stimulated PE-labelled B cells and organoids. Co-cultures were imaged using confocal microscopy at different timepoints during 48 h. We observed that B cells settled on the surface of the kidney organoid within 2 h. At 24 h, the morphology of the organoid was altered, and, at 48 h, it was fully distorted (Figure 5C).

To study whether DNA4 and DNA6-CAART could mitigate renal damage caused by IFNα-stimulated B cells, we added FITC-labelled DNA-CAART to the 3D immuno-kidney model 2 h following the organoid B cell co-culture. We observed that FITC-labelled DNA4-CAART killed PE-labelled B cells while preserving the structure of the kidney organoid (Figure 5A). Morphologic analysis showed that the kidney organoid maintained its circularity, aspect ratio, roundness, and solidity over time compared to the untreated condition, as well as when co-cultured with mock-transduced T cells (Figure 6B and Appendix A). However, DNA6-CAART treatment only produced an improvement in aspect ratio after 24 h of treatment compared to the control (0.173 vs. 0.996, *p* < 0.01, Figure 6B). No changes in circularity, solidity, or roundness were observed at the different time points.

Next, we evaluated the effect of DNA4 and DNA6-CAART therapy on kidney organoid cell death, injury and fibrosis formation by immunofluorescence at 48 h of incubation. DNA4-CAART treatment reduced apoptosis as measured by TUNEL assay and KIM-1 protein expression compared to the untreated 3D immuno-kidney model (fold decreases of 0.298 and 0.263, respectively, Figure 6C). DNA4-CAART reduced vimentin and collagen IV production significantly compared to the control (*p* = 0.001 and 0.033, respectively, Figure 6D). No changes were observed in apoptosis or KIM-1 expression with DNA6-CAART. In addition, DNA6-CAART only reduced vimentin formation but not collagen IV (*p* = 0.001).

## 3. Discussion

SLE is a heterogeneous disease characterised by autoantibody production and dysregulation of the immune system. In this study, we developed a novel precision cellular immunotherapy for anti-dsDNA-specific B cell depletion. We designed six DNA chimeric autoantibody receptors (DNA1-6-CAARs) containing extracellular antigen-binding domain sequences of α-actinin (DNA1-2), heparan sulphate (DNA3-4), histone-1 (DNA5), or C1q protein (DNA6). These proteins cross-react with anti-dsDNA, providing CAAR-engineered T cells with specific cytotoxicity for autoantibody-producing B cells. We have demonstrated in vitro that DNA4-CAART has the highest specific cytotoxicity, with lower renal damage as assessed by an immuno-kidney organoid model.

As CAR-T cell therapy continues to advance in the treatment of hematologic conditions, solid tumours, and autoimmune diseases, there is a growing interest in antigen-specific B cell depletion therapies. To date, four preclinical studies have used CAART cell therapy for autoimmune disorders: specifically for the treatment of pemphigus vulgaris [26], MuSK autoantibody-positive myasthenia gravis [27], NMDAR encephalitis [28], and experimental autoimmune encephalomyelitis (EAE) [29]. Traditional CARs incorporate a cleaved fragment of a monoclonal antibody known as a single-chain variable fragment (scFv) in the extracellular domain, whereas chimeric autoantibody receptors (CAARs) replace it with a specific antigen designed for autoantibodies [19]. Anti-dsDNA antibodies are specific to SLE, and serum levels often reflect disease activity in patients with LN [37]. Evidence supporting their role in disease pathogenesis comes mainly from animal [44] and human studies [45]. CAART cell therapy for LN would be an antigen tailored for anti-dsDNA antibodies. However, it remains unclear whether DNA or nucleosomes are the actual target of anti-dsDNA antibodies, or if there are other antigens, either self or foreign [46]. Several cross-antigens that mediate anti-dsDNA binding have been identified, such as histone-1 [47], C1q [48], α-actinin [49], and heparan sulphate [50]. Since DNA is broadly distributed in healthy and pathogenic states, we used antigens known to cross-link with anti-dsDNA as extracellular domains in the CAART cells [51].

DNA1-4 and DNA6-CAARs demonstrated robust antigen surface expression in primary human T cells, whereas in DNA5-CAART cells, expression was almost non-existent (less than 10%). Data indicate that DNA4 and DNA6-CAART specifically targeted the pathogenic anti-dsDNA-producing B cell population. In the first instance, we demonstrated, in vitro, the capacity of IFNα-stimulated B cells from anti-dsDNA^+^ LN patients to produce high titres of anti-dsDNA antibodies compared to those from anti-dsDNA^−^ patients. The co-incubation of anti-DNA^+^ B cells with DNA4 and DNA6-CAART resulted in 78.9% and 82.5% B cell reductions, respectively. This specific cytotoxic effect was confirmed by a reduction in anti-dsDNA antibody levels in the culture medium, increased secretion of interferon-gamma (IFNγ), and by an immunofluorescence and gene expression analysis of apoptosis-related genes. Conversely, DNA2 and DNA3-CAARTs exhibited minimal cytolysis, and DNA1-CAART produced significant non-specific lysis. Based on these results, we chose DNA4 and DNA6 for further study. These two constructs have shown the most promising results in terms of selective cytotoxicity, likely due to their interaction with anti-dsDNA antibodies. It has been previously reported that heparan sulphate can inhibit the binding of anti-DNA antibodies to DNA and may serve as a target antigen in vivo for cross-reactive anti-DNA antibodies [52,53]. Additionally, there is evidence suggesting that both mouse and human anti-DNA antibodies exhibit specific binding with the C1q protein [48]. We further investigated cellular interaction by confocal imaging, showing that the nature of the extracellular antigen domain influences the target-effector contact cell and could be relevant for its therapeutic effect. DNA4-CAART demonstrated major cell-to-cell contact after 2 h, whereas with DNA6-CAART, contact was observed after 6 h of co-incubation. Timing differences result from the different affinity and avidity of the binding interactions of the antigens [54]. SLE patients have serum anti-dsDNA that can neutralize or stimulate DNA-CAARTs. Preclinical studies on MuSK-CAART for myasthenia gravis treatment [27] and desmoglein 3 (DSG3)-CAART for mucosal pemphigus vulgaris [26] have shown that both CAARTs retained efficient levels of cytolysis activity in the presence of soluble antibodies. In line with previous studies, we showed, in vitro, that the specific cytotoxicity of DNA4 and DNA6-CAART remained similar in the presence of soluble anti-dsDNA antibodies, although it increased with a longer co-incubation time and higher effector-to-target ratio.

The use of three-dimensional immune organoids provides a highly representative in vitro microenvironment for studying immune-related disorders, enabling the interaction of immune cells with tissue structures and functions [55]. In oncology, co-culturing tumour organoids with homologous peripheral blood mononuclear cells (PBMCs) has been a valuable approach for investigating T cell activation and tumour cell apoptosis [56]. This methodology has also been applied to assess personalised CAR-T cell therapy [57]. In our study, we co-cultured kidney organoids with IFNα-stimulated anti-dsDNA-producing B cells from LN patients. The interaction of B cells with kidney organoids caused disruptions in their normal growth pattern. The therapeutic effects of DNA4-CAART cells in the immune kidney model were evaluated, demonstrating improvements in organoid morphology, a reduction in organoid cell death, and renal damage markers such kidney injury molecule-1 (KIM-1). KIM-1 expression in renal biopsies is known to predict ongoing glomerular and tubulointerstitial inflammation [58]. The formation of fibrosis in kidney tissue is characterised by the excessive deposition of extracellular matrix components such as collagen IV [59] and vimentin [60]. During the progression of fibrosis, immune cell-mediated inflammation in the kidney activates intrinsic renal cells, leading to the production and release of profibrotic cytokines and growth factors [61]. Considering that CAART cells significantly reduce the effects of B cells, we assessed their potential to attenuate fibrosis formation. Our findings indicate that DNA4-CAART markedly reduce vimentin and collagen IV in immune kidney organoids. While DNA6-CAART only reduce vimentin, no discernible improvement in kidney organoid morphology was observed. Data suggest that DNA4-CAART exhibit superior therapeutic effects directly on kidney tissue, possibly due to their higher cell-to-cell contact kinetics. Despite the promising applications of organoid technology, our immune organoid system has several important limitations. These include the absence of a vascular system and the use of Matrigel in the organoid culture medium, which could potentially affect the viability of immune cell cultures and their interaction with kidney organoids. Additionally, a limitation of our study is that the kidney organoids lacked podocytes and endothelial cells, presenting only epithelial and proximal tubule cells. Therefore, conducting in vivo experiments will be necessary to further explore the potential of DNA-CAART therapy.

## 4. Materials and Methods

### 4.1. Design and Construction of CAAR Vectors

To engineer T cells to kill anti-dsDNA-producing B cells in LN, we used a modified lentiviral plasmid, Lenti-ONE CAR (GEG Tech, Villebon-sur-Yvette, France), to express a chimeric autoantibody receptor (CAAR) as CAAR extracellular domains with antigens are known to cross-react with anti-dsDNA antibodies [62].

We selected six different human sequences to obtain CAAR vectors: two sequences based on alpha-actinin (GenBank ACJ24535.1, N-ter and C-ter sequence), two of heparan sulphate proteoglycan 2 (GenBank KAI2515460.1, N-ter and C-ter sequence), one of histone 1 (H1) sequence (epitope AK PKTAKPKAAK PKKAAAKK), and one of the C1q sequence (epitope A08, C1q A15-27: GRPGRRGRPGLKG) [62,63]. All CAAR designs include a linker of 20 amino acids to optimize construction and activity. Extracellular domains were fused to the activator domains CD28-CD3ζ, which have been used successfully in CD19 CAR clinical trials [64]. All these open reading frame (ORF) vectors are under the control of a hEF1α promoter optimised by GEG Tech (Paris, France, Appendix A).

### 4.2. Patient Characteristics

SLE patients with previous history of biopsy-proven proliferative LN (type III or IV ± V), recruited from the Lupus Unit at Vall d’Hebron Hospital, and healthy donors as the control group, were included (N = 10 for each group). SLE patients were classified into two groups according to their persistent serum anti-dsDNA positivity (anti-dsDNA^+^ or anti-dsDNA^−^) at inclusion (Table 1). Patients in both groups had a mean follow-up of 3.7 years since renal biopsy (range 2.5–4.9 years). The study was approved by the ethics review committee at Vall d’Hebron Hospital (PI18/01917, 2 March 2018). Written informed consent was obtained from participants before study inclusion.

### 4.3. In Vitro Transduction and Expansion of CAART Cells

Primary human T cells were isolated from patients’ PBMCs using “Dynabeads Untouched Human T Cells Kit” (ThermoFisher Scientific, Waltham, MA, USA) following the manufacturer’s instructions (more details in Appendix A). They were cultured in six-well culture plates coated with RetroNection (9 ng/mL, Takara Bio Europe, Saint-Germain-en-Laye, France) and RPMI-supplemented medium prior to transfection with CAAR or control constructs. After 24 h, each lentiviral CAAR particle was added (2–10 µL) and cells were expanded by static culture for 2–3 days [65]. Fresh medium was added containing IL-2 (20 ng/Ml, ThermoFisher Scientific, Waltham, MA, USA). T cell expansion was monitored for 5–7 days by the measurement of cell volume and concentration (Coulter counter, Beckman Coulter, Brea, CA, USA). Cell surface expressions of the CAAR constructs were validated and quantified by flow cytometry (BD LSRII), using primary antibodies, and by qPCR-RT analysis.

### 4.4. In Vitro Cytotoxicity Assays

Primary human B cells from the study groups (anti-dsDNA^+/−^ antibodies) were isolated using “Dynabeads Untouched Human B cells Kit” (Invitrogen, Waltham, MA, USA) following the manufacturer’s instructions and stimulated with IFNα (more details in Appendix A). The in vitro killing of isolated IFNα-stimulated B cells from the study groups (LN anti-dsDNA^+/−^) by CAART cells was assessed by flow cytometry and immunofluorescence. CAART or non-transduced CAART (control) cells were co-cultured with IFNα-stimulated B cells in 24-well plates for 24 h at various effector–target (E:T) ratios. Next, we used the “eBioscience Annexin V apoptosis” kit (Invitrogen, Waltham, MA, USA) following the manufacturer’s instructions to evaluate apoptosis by flow cytometry (BD LSRII). Co-culture supernatants were harvested after completing the final plate reading at 4 h and stored at −80 °C for IFNγ measurement by ELISA. All experiments were performed with at least 3 replicates and with primary human T cells from LN patients (n = 5 in each group).

The specific lysis of B cells by CAART cells was also analysed using immunofluorescence. B cells were stained using anti-human CD19-PE (Invitrogen, Waltham, MA, USA) for 30 min at room temperature. After staining, they were cultured with CAART cells at several concentrations (E:T ratios of 1, 2.5, 5, 10, or 20) during 24 h. CD19+ cell viability was analysed using fluorescence microscopy (Olympus BX61) and Image J software version 1.51.

### 4.5. Confocal Live-Cell Microscopy Interaction between B Cells and CAART Cells

Isolated study B cells were stimulated with IFNα (50 ng/mL, Gibco, Thermofisher Scientific, Waltham, MA, USA) and stained with human CD19-PE (Invitrogen, Waltham, MA, USA). CAART cells were stained using human primary antibody-FITC (Appendix A) prior to starting the confocal live-cell experiments. After that, they were stained with DAPI and co-cultured at an E:T ratio of 10. To study the interaction, images of the selected sections were taken at 2, 6, and 24 h using a Zeiss LSM780 confocal microscope. Fluorescence (488 nm and 580 nm) and bright-field micrographs of captured cells were obtained at 1024 × 1024, 16-bit (×25), and analysed using Image J software version 1.51.

### 4.6. RNA Extraction and RT-qPCR

Total RNA from isolated B or T cells was extracted using the miRNeasy^®^ Mini Kit (Qiagen, Hilden, Germany) according to the manufacturer’s instructions. RT-qPCR was performed on a 7000 ABI Thermofisher (Applied Biosystems, Waltham, MA, USA) using a TaqMan gene expression assay (FAM dye labelled MGB probe (Applied Biosystems, Waltham, MA, USA)), and the gene-specific primers and probes are shown in Appendix A (more details in Appendix A).

### 4.7. Kidney Organoids B Cell Co-Culture to Evaluate Therapeutic CAART Cell Efficacy

Kidney organoids were purchased from Hubrecht Organoid Technology (Utrecht, The Netherlands, passage 4–6). Kidney organoid cultures were established from frozen stocks following the manufacturer’s instructions (more details in Appendix A). Organoids were maintained in an organoid medium in 6-well culture plates (suspension surface, Sarstedt, Germany) coated with 70–80% of Matrigel (Corning, New York, NY, USA) and Cultrex Basal Membrane Extracts (BME, Sigma-aldrich, San Luis, MI, USA) mix (1:1) in a 37 °C incubator at 5% CO_2_. For further experiments, organoids were seeded into 24-well plates using two 10 uL drops (1000 cells/cm^2^). They were co-cultured with 10^3^ IFNα-stimulated primary B cells isolated from the study groups and healthy donors for 2 days. Next, 10^4^ CAART cells were added into the culture and after 2, 24, and 48 h, their effect on the kidney organoids was evaluated using a confocal microscope (Zeiss LSM980). To measure the surface area of the organoid, bright-field images were captured every 10 μm in the Z-direction from the top to the bottom of the organoid. Ten organoids were used for surface area measurements using Image J software version 1.51. Organoids without co-culture were used as the control.

### 4.8. Immunofluorescence Analysis of Kidney Organoids after Exposure to CAART Cells

Kidney organoids exposed to IFNα-stimulated B cells and CAART cells were fixed in 4% paraformaldehyde-PBS and were paraffin-embedded. Immunofluorescence was performed using standard procedures, including heat-induced antigen retrieval. DAPI was used for nuclear staining. Overnight incubation of primary antibodies was performed at 4 °C. Rabbit anti-human kidney injury molecule-1 (KIM-1, Thermofisher Scientific, Waltham, MA, USA), rabbit anti-human collagen IV (Thermofisher Scientific, Waltham, MA, USA), and mouse anti-human vimentin (Thermofisher Scientific, Waltham, MA, USA, Appendix A) were used following the manufacturer’s instructions. Afterwards, anti-rabbit or anti-mouse secondary antibodies were incubated for 2 h at room temperature (Appendix A). Cell death was measured using the Click-iT Plus TUNEL assay kit (Invitrogen, Waltham, MA, USA). Immunofluorescence staining was imaged on a Zeiss LSM710 confocal microscope. Quantification was performed on ≥10 organoids/sample using Image J software version 1.51.

### 4.9. Enzyme-Linked Immunosorbent Assay (ELISA)

Patients’ serum samples and cell culture media were used for antibody quantification. Anti-dsDNA and anti-C1q antibody titres were measured by commercial ELISA (Cusabio technology LLC, Houston, TX, USA and INOVA Diagnostics Inc., San Diego, CA, USA, respectively) according to the manufacturer’s instructions. In-house ELISA was optimised to detect positive patients for anti-α-actinin, anti-histone1, and anti-heparan sulphate following the described methodology (more details in Appendix A) [66]. Interferon-γ production was quantified in the supernatants of the 48 h effector and target cells’ co-culture in 24-well culture plates using the “IFN gamma Human Elisa Kit” (Thermofisher Scientific, Waltham, MA, USA) according to the manufacturer’s recommendations.

### 4.10. Statistical Analysis

Statistical analysis was performed using GraphPad Prism software version 6.0.1. The results are presented as means ± SEM and were analysed using one-way ANOVA followed by Tukey’s test to detect differences between groups. Survival curves were compared via a Mantel–Cox log-rank test. Categorical variables were expressed as counts and a proportion of patients (%) and compared using Fisher’s exact test. A *p* value < 0.05 was considered statistically significant for each of the experiments.

## 5. Conclusions

In summary, our study demonstrates the promising potential of DNA-CAART as a novel precision cellular immunotherapy for the depletion of anti-dsDNA-specific B cells in the treatment of lupus nephritis. Our future research direction will focus on further exploring the efficacy, in vivo, of our most promising CAART formulation, DNA4-CAART, using experimental animal models.

## Figures and Tables

**Figure 1 ijms-25-04226-f001:**
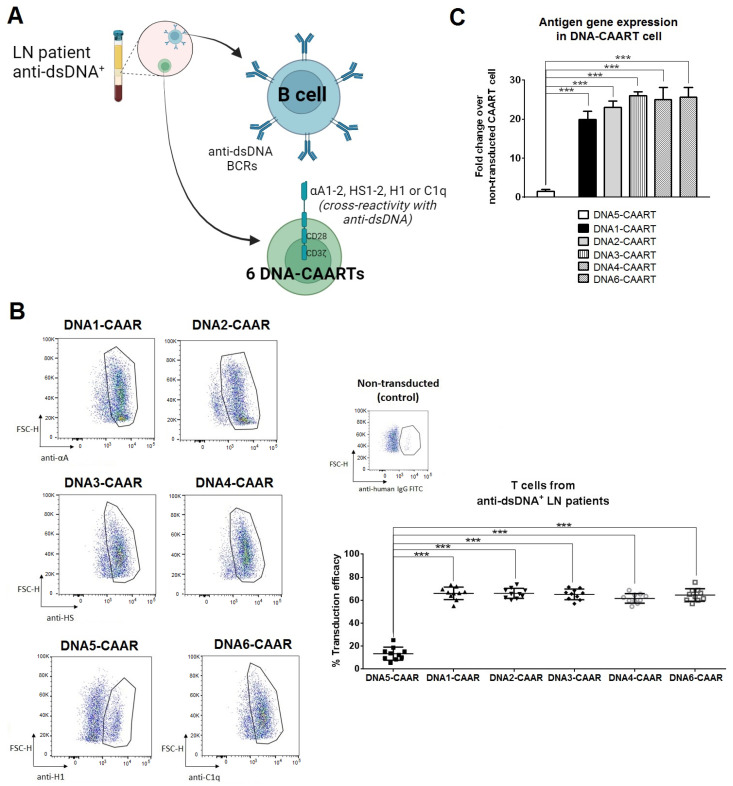
DNA-CAART cell expression on primary human T cells from anti-dsDNA^+^ LN patients. (**A**) Schematic concept of the proposed DNA-CAART therapy for LN treatment. Six DNA-CAARTs were designed using different autoantigen protein sequences: DNA1 and DNA2-CAARs with sequence for α–actinin protein (αA1-2); DNA2 and DNA3-CAAR with sequence for heparan sulphate (HS1-2); DNA5-CAAR sequence for histone-1 (H1) and DNA6-CAAR sequence for C1q (C1q). Cross-reactivity of these antigens with BCRs for anti-dsDNA in B cells will induce their selective B cell depletion. Figure made in BioRender.com (accessed on 16 January 2024). (**B**) Primary human T cells were transduced with DNA1-6-CAAR lentivirus and CAAR expression was detected using anti-protein FITC antibody. CAAR^+^ transduction efficiency was determined in anti-dsDNA^+^ LN patients (N = 10). *** *p* < 0.001, one-way analysis of variance (ANOVA) was performed to analyse differences between six groups and *t*-test between two groups. (**C**) Gene expression of CAAR antigens (alpha-actinin, heparan sulphate, histone 1, and C1q) was evaluated in each DNA1-6-CAART cell from samples of anti-DNA^+^ LN patients (N = 10). Fold changes were calculated over control (non-transduced T cells). *** *p* < 0.001, one-way analysis of variance (ANOVA) was performed to analyse differences between six groups.

**Figure 2 ijms-25-04226-f002:**
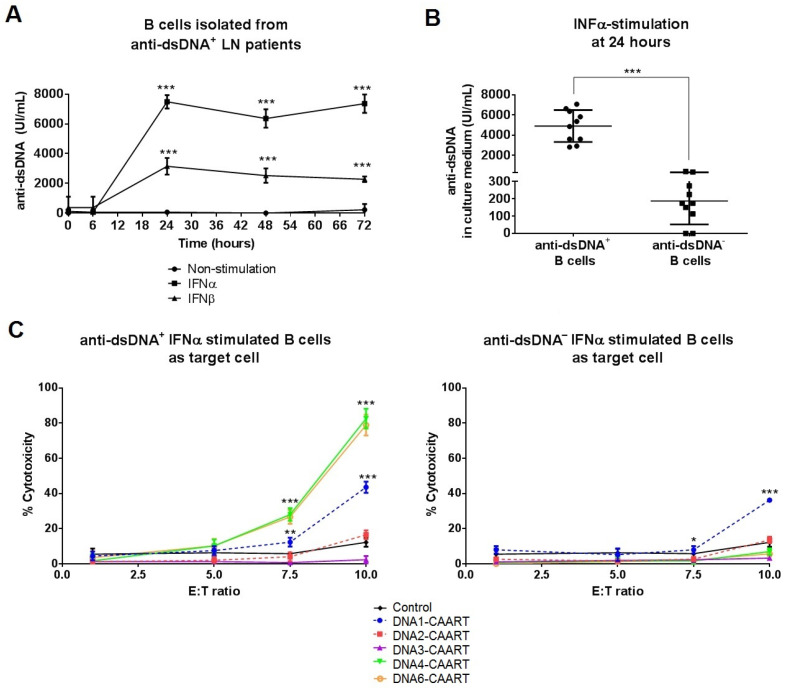
Evaluation of selective anti-DNA^+^ B cell depletion by DNA-CAART cells. (**A**) In vitro anti-dsDNA production of B cells isolated from anti-DNA^+^ LN patients (N = 10) was evaluated after being stimulated with IFNα or IFNβ (50 ng/mL) or in non-stimulated conditions (PBS buffer) for 72 h. One-way analysis of variance (ANOVA) was performed to analyse differences between three groups. *** *p* < 0.001. (**B**) Anti-dsDNA levels of IFNα-stimulated B cells isolated from anti-dsDNA^+^ or anti-dsDNA^−^ LN patients (N = 10 each group). Levels were measured in culture medium after 24 h of stimulation by ELISA. Student’s *t*-test. *** *p* < 0.001. (**C**) Cytotoxicity by DNA-CAAR T cells (effector cells) against anti-dsDNA^+^ or anti-dsDNA^−^-producing B cells (target cells) were measured at different E:T ratios after 24 h of co-culture. The percentage of lysed B cells was analysed by flow cytometry. Error bars represent the mean ± SEM from culture experiments (N = 10). Significant differences were calculated in comparison with the control (non-transduced T cells) in each E:T ratio using a paired *t*-test. * *p* < 0.05, ** *p* < 0.005 and *** *p* < 0.001.

**Figure 3 ijms-25-04226-f003:**
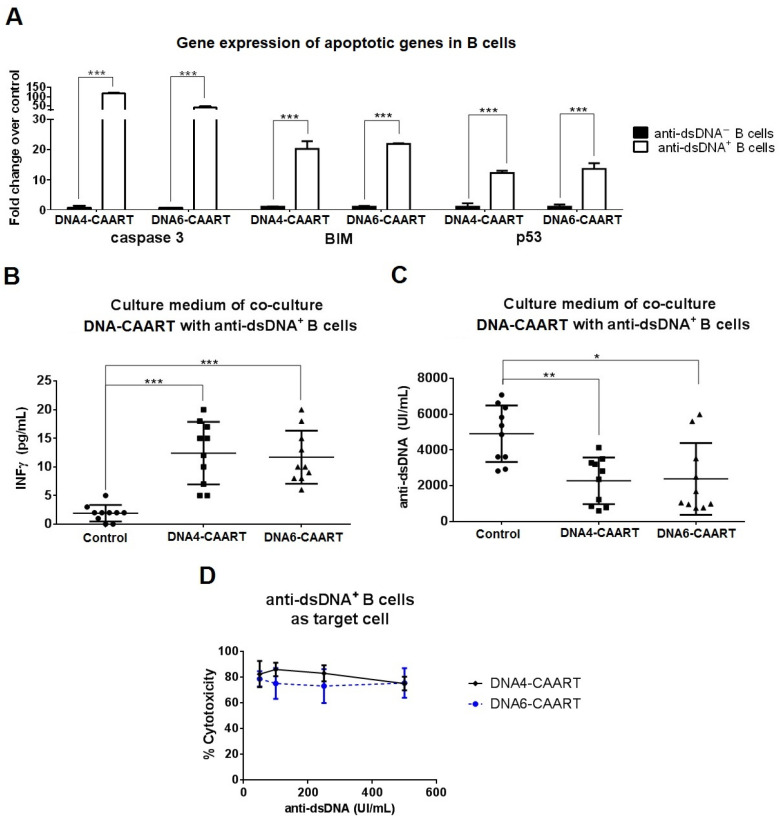
Evaluation of selective anti-DNA^+^ B cells depletion by DNA-CAAR T cells. (**A**) Gene expressions of apoptotic genes (caspase 3, BIM, p53) were evaluated in DNA-CAAR T cell co-cultures from samples of anti-dsDNA^+^ or anti-dsDNA^−^ LN patients (N = 10). Fold changes were calculated over control (non-transduced T cells) by Student’s *t*-test. *** *p* < 0.001. (**B**,**C**) Enzyme-linked immunosorbent assay (ELISA) was performed in supernatants of DNA-CAAR T cells co-incubated with anti-dsDNA^+^ B cells (10:1 E:T, 24 h, experiments run using primary cells from different LN patients (N = 10)) to measure IFNγ (**B**) or anti-dsDNA levels (**C**). One-way analysis of variance (ANOVA) was performed for multiple comparisons. * *p* < 0.05, ** *p* < 0.005 and *** *p* < 0.001. (**D**) Standards of anti-dsDNA antibodies (0, 50, 100, 250, 500 UI/mL) were added to the co-culture of DNA4 or DNA6-CAART with IFN-α-stimulated anti-dsDNA^+^ B cells (ratio 10:1, E:T). After 24 h, the cytotoxicity was not altered in these conditions.

**Figure 4 ijms-25-04226-f004:**
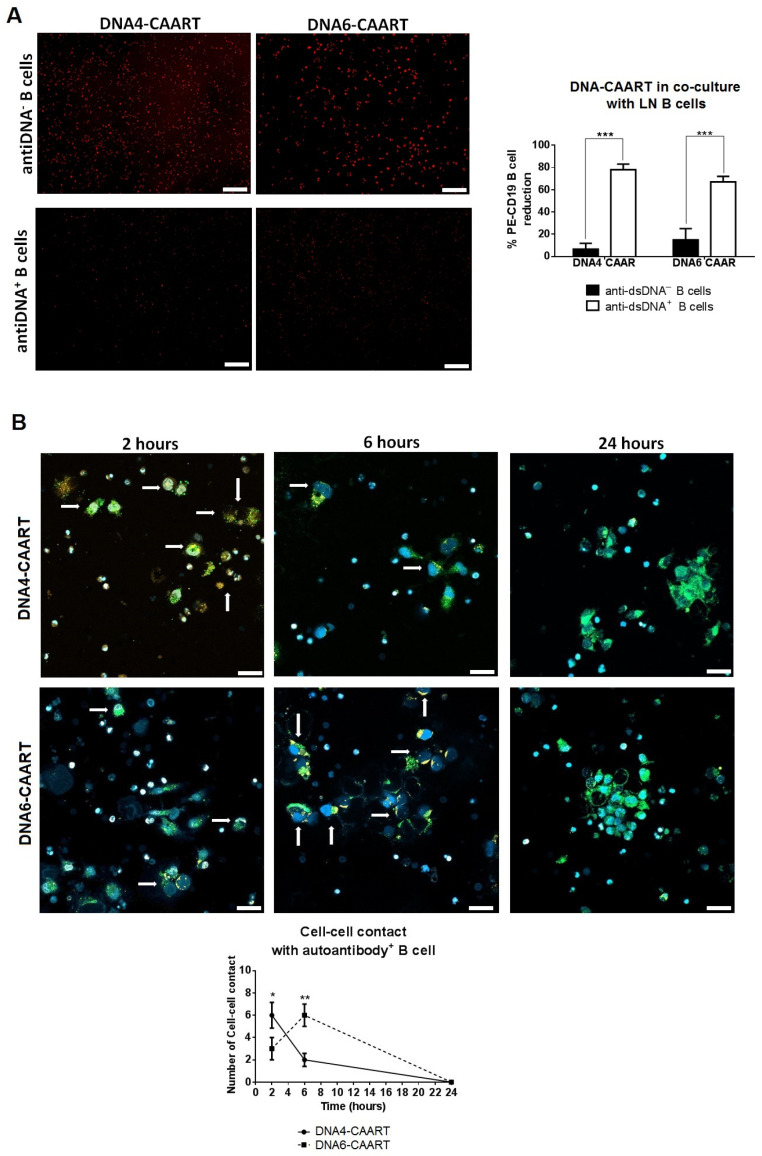
Visualization and evaluation of DNA4 and DNA6-CAART interaction with target B cells. (**A**) In vitro co-culture of PE-CD19 labelled B cells isolated from anti-dsDNA^+/−^ LN patients with DNA4 or DNA6-CAART (n = 10). Immunofluorescence images were captured after 24 h (ratio E:T, 10:1). Scale bar = 2 µm. One-way analysis of variance (ANOVA) was performed to analyse differences between three groups. *** *p* < 0.001. (**B**) Confocal imaging visualised DNA4 or DNA6-CAART (green)-mediated interaction of anti-dsDNA^+^ B cell (red) within 6 h of cell–cell contact. DNA-CAART were co-cultured with anti-dsDNA^+^ B cells in a effector-to-target ratio of 10:1. Image series illustrates the dynamics of DNA-CAART killing at 2, 6, and 24 h. Cell–cell contacts are marked with white arrow. * *p* < 0.05 and ** *p* < 0.005. Scale bar = 50 µm.

**Figure 5 ijms-25-04226-f005:**
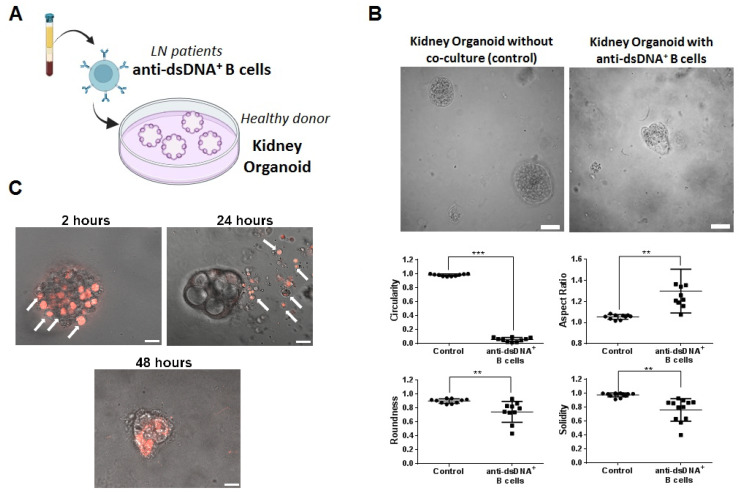
B cells from anti-DNA^+^ patients damage kidney organoid overtime. (**A**) Scheme of co-culture between anti-dsDNA^+^ B cells from LN patients and kidney organoid established from healthy donor cells. (**B**) Study of kidney organoid morphology after co-culture with anti-dsDNA+ IFNα-stimulated B cells for 48 h (n = 5). Scale bar = 50 µm. One-way analysis of variance (ANOVA) was performed to analyse differences between the groups. ** *p* < 0.005, *** *p* < 0.001. (**C**) Immunofluorescence image captured at 2, 4, 6, 18, 24, and 48 h using confocal microscopy to characterize the interaction between anti-dsDNA+ IFNα-stimulated B cells and kidney organoids. Arrows mark the positive PE-labelled anti-dsDNA^+^ B cells (red). Scale bar = 100 µm.

**Figure 6 ijms-25-04226-f006:**
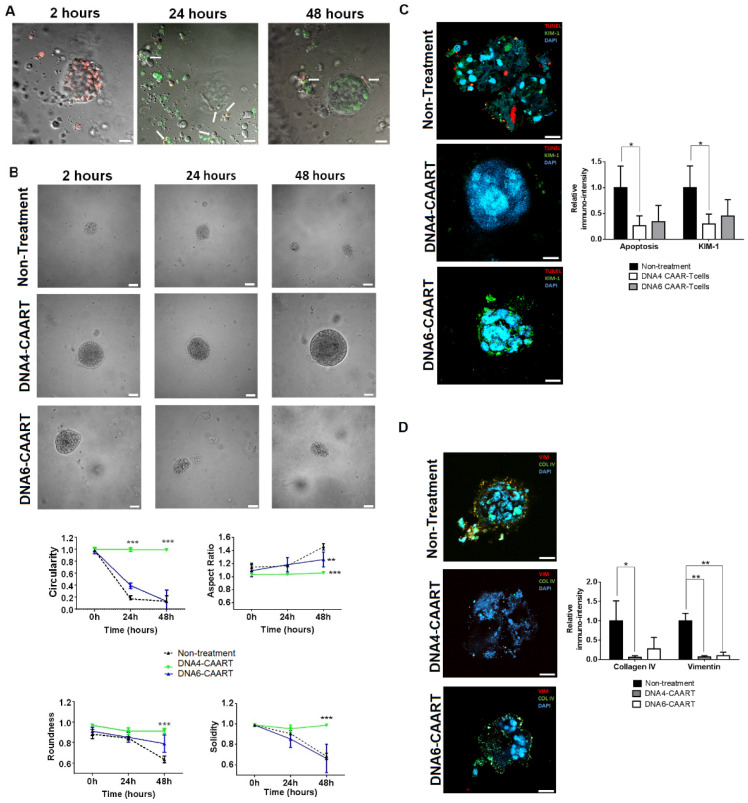
Therapeutic effect of DNA4 and DNA6-CAART on 3D immune-kidney organoid model damage. (**A**) In vivo imaging of 3D immune-kidney organoid with DNA4-CAART, where B cells were labelled with PE (red) and DNA4-CAART with FITC (green) two hours after the B cell-organoid co-culture DNA-CAART was added. Arrows indicate the killing of B cells by DNA4-CAART at 24 h and 48 h. Scale bar = 50 µm. (**B**) Morphology analysis of 3D immune-kidney organoid after 2, 24 and 48 h incubation with DNA4 or DNA6-CAART treatment or without treatment. Analyses of circularity, aspect ratio, roundness, and solidity were performed using Image J software version 1.51 (n = 5). Scale bar = 100 µm. Statistical analysis was performed between groups using two-way ANOVA. Statistical significance shown is in comparison with untreated control. ** *p* < 0.005, *** *p* < 0.001. (**C**) TUNEL assay was performed to quantify apoptotic cells (red) and protein levels of KIM-1 were quantified by immunofluorescence (green). Relative intensity of the fluorescence value is used as non-treatment conditions (control), and absolute values were obtained using Image J software version 1.51 (n = 5). Scale bar = 20 µm. One-way ANOVA was performed to analyse differences between the three groups. * *p* < 0.05. (**D**) To evaluate fibrosis, collagen IV and vimentin levels were measured and analysed via a one-way ANOVA test to obtain statistical significance between groups. Scale bar = 20 µm. * *p* < 0.05, ** *p* < 0.005.

**Table 1 ijms-25-04226-t001:** Baseline demographic and disease characteristics of the study groups at inclusion.

	Lupus Nephritis Anti-DNA^+^ (n = 10)	Lupus Nephritis Anti-DNA^−^ (n = 10)	*p* Value ^a^
** *Demographic* **			
**Age (years)**	41.2 ± 8.3	42.8 ± 12.8	0.744
**Gender (Female/male)**	10/10	8/10	0.136
** *Race/ethnicity, n (%)* **			
**White**	4 (40)	6 (60)	0.371
**Hispanic**	6 (60)	4 (40)	
**Other**	0 (0)	0 (0)	
** *Laboratory parameters* **			
**Serum creatinine, mg/dL**	1.1 ± 0.9	0.8 ± 0.2	0.317
**eGFR (mL/min)**	87 ± 6.6	88 ± 3.4	0.675
**Serum C3, mg/dL**	75 ± 32	96 ± 16	0.08
**Serum C4, mg/dL**	13.9 ± 8.8	16.4 ± 11.9	0.600
**Hypocomplementemia**	6 (60)	3 (30)	0.178
**Proteinuria, g/24 h**	1.9 ± 1.5	1.3 ± 1.5	0.383
** *Autoantibody status at inclusion* **			
**ANA titre ≥ 1/80, n (%)**	10 (100)	8 (80)	0.474
**Anti-dsDNA antibodies ≥ 15 IU/mL, n (%)**	10 (100)	0 (0)	<0.0001
**Anti-dsDNA antibodies, IU/mL**	268.5 ± 237.6	2.8 ± 8.8	0.0024
**Anti-SSA/Ro antibodies, n (%)**	6 (60)	3 (30)	0.177
**Anti-SSA/La antibodies, n (%)**	2 (20)	0 (0)	0.136
**Anti-RNP antibodies, n (%)**	4 (40)	4 (40)	1.000
**Anti-Sm antibodies, n (%)**	6 (60)	4 (40)	0.371
**Anti-C1q, n (%)**	6 (60)	0 (0)	0.011
**Anti-Histone1, n (%)**	6 (60)	1 (10)	0.057
**Anti-α-actinin, n (%)**	6 (60)	1 (10)	0.057
**Anti-heparan sulphate, n (%)**	9 (90)	1 (10)	0.0011
** *Disease index (SLEDAI-2K)* **			
**Total score**	5.8 ± 1.8	4.4 ± 2.0	0.117
** *Treatment at inclusion* **			
**Antimalarial agents, n (%)**	7 (70)	4 (40)	0.177
**Immunosuppressive therapy, n (%)**	10 (100)	10 (100)	1.000
**MMF/EC-MPS**	10 (100)	10 (100)	1.000
**Anticalcineurinic agents**	2 (20)	2 (20)	1.000
**Corticosteroid use, n (%)**	8 (80)	9 (90)	0.531
**Daily prednisone dose, mg/day**	2.5 ± 0.8	2.0 ± 0.5	0.111
** *Type of GMN at flare, n (%)* **			
**Class III**	3 (30)	2 (20)	1.000
**Class IV**	5 (50)	6 (60)	1.000
**Class III/IV-V**	2 (20)	2 (20)	1.000
**Activity Index, mean**	6.3 ± 5.2	3.3 ± 1.9	0.104
**Chronicity Index, mean**	3.7 ± 1.7	2.8 ± 1.0	0.166
** *Autoantibody status at renal biopsy* **			
**ANA titre ≥ 1/80, n (%)**	10 (100)	10 (100)	1.000
**Anti-dsDNA antibodies ≥ 15 IU/mL, n (%)**	10 (100)	7 (70)	0.210
**Anti-dsDNA antibodies, IU/mL**	384.6 ± 165.9	179.14 ± 297.35	0.073
**Anti-SSA/Ro antibodies, n (%)**	6 (60)	3 (30)	0.177
**Anti-SSA/La antibodies, n (%)**	2 (20)	0 (0)	0.136
**Anti-RNP antibodies, n (%)**	4 (40)	4 (40)	1.000
**Anti-Sm antibodies, n (%)**	6 (60)	4 (40)	0.371
**Anti-C1q, n (%)**	6 (60)	2 (20)	0.169
**Anti-Histone1, n (%)**	5 (50)	2 (20)	0.349
**Anti-α-actinin, n (%)**	7 (70)	5 (50)	0.649
**Anti-heparan sulphate, n (%)**	10 (100)	7 (70)	0.210

Values are expressed as mean ± standard deviation (SD). eGFR, estimated glomerular filtration rate; anti-dsDNA, anti-double-stranded DNA (reference range < 35 UI/mL); SLEDAI-2K: Systemic Lupus Erythematosus Disease Activity Index 2000 [42]; MMF: mycophenolate mofetil; EC-MPS: enteric-coated mycophenolate sodium; Reference ranges are as follows: anti-double-stranded DNA antibodies < 27 IU per millilitre; anti-SSA/Ro, anti-SSA/La, anti-RNP, and anti-Smith < 20 chemiluminescent units (CU); anti-C1q/anti-histone1/anti-α-actinin and anti-heparan sulphate  < 25 IU per millilitre; ^a^
*p*-value refers to the comparison of LN anti-DNA^+^ with LN anti-DNA^−^ patients using the ANOVA *t*-test or Fisher’s exact test.

## Data Availability

Data are contained within the article and Appendix A.

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
