# Peer review of "Precise Targeting of Autoantigen-Specific B Cells in Lupus Nephritis with Chimeric Autoantibody Receptor T Cells"

_ijms, 2024, doi:10.3390/ijms25084226_

Round 1

Reviewer 1 Report

Comments and Suggestions for Authors

Comments on the Quality of English Language

Minor editing of Enghish language is required

Reviewer 2 Report

Comments and Suggestions for Authors

The work presented by Sole et al is very interesting and there are only a few comments:

- The authors do not elaborate on why 2 of the constructs work well while the others do not. If possible they should mention their thoughts on this.

- The kidney organoid is not clearly described, for instance which cells are present in the organoid and has it been validated.

- What is meant by aspect ration in line 277?

minor:

In Fig 4a it is difficult to see is there is staining.

typo's: 

the authors use both IFN and INF in the manuscript, it should be IFN

Line 108 is double, line 211 citotixicity, line 352 miastenia gravis, Fig 3A gens 

Comments on the Quality of English Language

The English is sufficient, see remarks on typo's.

Reviewer 3 Report

Comments and Suggestions for Authors

This paper describes the generation and in vitro testing of chimeric autoantibody receptor T cells directed against dsDNA-specific B cells, with the potential future aim of employing such cells for systemic lupus erythematosus treatment and for preventing the development of lupus nephritis.

The authors engineer T-cell activating constructs by fusing antigens cross-reactive to dsDNA with intracellular signaling chains. Five of six constructs are efficiently expressed by transduced blood T cells and two of them exert cytotoxicity on target B cells isolated from seropositive but not seronegative patients. Target killing is accompanied by IFNg production, expression of pro-apoptotic genes and reduced anti-dsDNA antibody production in the cultures. In an organoid-immune co-culture assay CAAR-T cells protect kidney organoids from detrimental effects of activated B cells.

Minor remarks

- The sentence in lines 48-49 is confusing, belimumab is not a survival factor. It is suggested that therapeutic antibody names and targets are consistently identified in the introduction.

- Line 60 “In SLE, studies support the efficacy of CD19 CAR-T cell therapy by inducing persistent CD19 B cell depletion, reduction of autoantibody production and clinical remission [21-24].”
The cited studies are partly murine models, so this sentence is misleading. It is suggested that murine models are mentioned and referred to separately.

- There are several misspelt words throughout the text; examples : Figure 3 “Apoptotic gens”, “visualitzation of DNA4” line 218, “confocal microscopy” line 256. Please run a spell check to correct such words.

Major remark

The experiment with kidney organoids is meant to prove that the CAAR-T cells can provide protection against B-cell mediated damage via the mechanism of action described in the paper. While it would seem logical that the transduced T cells achieve organoid protection by killing dsDNA-specific B cells, the experiment does not prove it. What is shown is that interferon alpha primed B cells from seropositive patients mediate damage to the organoid, which damage is weakened in the presence of transduced T cells. In the materials and methods section the authors state: “They [organoids] were co-cultured with 10^3 INFα-stimulated primary B cells isolated from the study groups and healthy donors for 2 days.” The reviewer could not find results with experiments using organoids and healthy donor B cells, which would rule out that IFN alpha stimulated B cells can interfere with organoid growth even when the B cells are not dsDNA specific. Another control with mock-transduced T cells or CAART with different specificity would be required to prove that specific interaction between CAART and dsDNA specific B cells is required for the observed effects. How do the authors think specificity in the chain of these events is proven without those controls?

Comments on the Quality of English Language

Spelling mistakes could be corrected by running language check in text editor.
